# Development of Ogura CMS Fertility-Restored Interspecific Hybrids for Use in Cytoplasm Replacement of Golden-Heart Materials in *Brassica rapa*

**DOI:** 10.3390/genes14081613

**Published:** 2023-08-11

**Authors:** Ze Li, Guoliang Li, Fei Li, Shifan Zhang, Xiaowu Wang, Jian Wu, Rifei Sun, Shujiang Zhang, Hui Zhang

**Affiliations:** 1Institute of Vegetables and Flowers, Chinese Academy of Agricultural Sciences, Beijing 100081, China; 13163135502@163.com (Z.L.); liguoliang@caas.cn (G.L.); lifei@caas.cn (F.L.); zhangshifan@caas.cn (S.Z.); wangxiaowu@caas.cn (X.W.); wujian@caas.cn (J.W.); sunrifei@caas.cn (R.S.); 2State Key Laboratory of Vegetable Biobreeding, Institute of Vegetables and Flowers, Chinese Academy of Agricultural Sciences, Beijing 100081, China

**Keywords:** Ogura CMS, restorer of fertility gene (*Rfo*), *GOLDEN* gene, *Brassica rapa*

## Abstract

Ogura cytoplasmic male sterility (CMS) is one of the important methods for hybrid seed production in cruciferous crops. The lack of a restorer of fertility gene (*Rfo*) in *Brassica rapa* L. restricts the development and utilization of its germplasm resources. In this research, *Brassica napus* with the *Rfo* gene was used to restore the fertility of Ogura CMS *B. rapa* with the golden heart trait. Through the distant cross of two *B. rapa* and four *B. napus*, six interspecific hybrid combinations received F1 seeds. The six combinations were different in seed receiving. By morphological observation and molecular marker-assisted selection (MAS), in F1, individuals containing the *Rfo* gene all appeared fertile, while those without it remained male-sterile. The pollen viability of the fertile individuals was measured, and the fertile lines of the six interspecific hybrid combinations were different (40.68–80.49%). Three individuals (containing both *Rfo* and *GOLDEN* genes) with the highest pollen vitality (≥60%) were backcrossed with fertile cytoplasmic *B. rapa,* resulting in a total of 800 plants. Based on the MAS, a total of 144 plants with *GOLDEN* but no *Rfo* were screened (18%). Moreover, through morphological investigation, one individual with normal cytoplasm, stable fertility but without the restoring gene *Rfo*, the *GOLDEN* gene, and morphological characteristics similar to those of *B. rapa* was obtained. These results increased the diversity of *B. rapa* germplasm and provided a new method for the utilization of CMS germplasm in *Brassica* crops.

## 1. Introduction

Male sterility is widespread in higher plants and has been reported in more than 600 plant species since the first case was reported in 1763 [1]. The term usually refers to the degeneration or loss of function of male gametes, but with the normal development of female gametes and normal fertilization and fruit-bearing with foreign pollen [2]. Male sterility can be divided into three types: genic male sterility (GMS), controlled by nuclear genes; cytoplasmic male sterility (CMS), controlled by cytoplasmic genes; and nucleus-cytoplasmic interaction male sterility (NCR), controlled by both nuclear genes and cytoplasm [3,4].

CMS is a maternal genetic phenomenon that results in abnormal fertilization and fruiting due to pollen abortion and is one of the most effective ways to use plant heterosis [2]. Therefore, there are currently many studies on the mechanisms of cytoplasmic male sterility and fertility restoration. Cytoplasmic male sterility (CMS) is jointly regulated by mitochondrial sterile genes and nuclear restorer of fertility (Rf) genes. Cytoplasmic mitochondrial gene products lead to male infertility, while nuclear fertility-restoring gene (Rf) products can alter the transcription or expression of sterile genes in mitochondria, leading to the restoration of fertility in sterile lines [5].

The types and mechanisms of cytoplasmic male sterility vary among different crops. Ogura CMS was the first natural male sterile type and was discovered by Ogura [6] in radish. In 1977, Bannerot introduced the nucleus of *B. napus* into the Ogura CMS of radish through interspecific hybridization and continuous backcrossing, obtaining the Ogura CMS of *B. napus* [7]. Bonhomme et al. [8,9] found a unique 2.5 kb segment of NcoI in Ogura CMS cytoplasm through mtDNA analysis. Through sequence analysis, it was found that this segment contains *orf138*, *orf158*, and a tRNA gene, which may be related to male infertility. *Orf138* is co-transcribed upstream of *orf158* (which encodes 8 subunits of ATP synthase), producing a 1.4 kb transcript [10]. Duroc et al. [11] found that in Ogura CMS, *orf138* encodes a membrane-binding protein that is linked to the inner mitochondrial membrane with a molecular weight of 22 kD and can easily form oligomers. ORF138 was transferred into E. coli for expression, and the growth of E. coli was inhibited, indicating that ORF138 was toxic to cells. By constructing a targeted mitochondrial expression vector fused with *orf138* and the green fluorescent protein GFP, the expression vector of this gene was introduced into yeast and Arabidopsis to study the mechanism of the relationship between the structure of sterile proteins and male sterility. It was found that the ORF138 protein did not inhibit the growth of yeast but changed the shape of mitochondria in yeast and plant cells, and transgenic Arabidopsis did not exhibit sterile traits [12]. This may be related to the subcellular localization of ORF138 expressed in the nucleus.

The study of restoration genes not only helps to shorten the breeding period, but also serves as an important basis for the classification of different types of CMS and the study of nuclear-cytoplasmic interactions. Due to the completion of genome sequencing in various crops, the localization and cloning of CMS recovery genes have made rapid progress. Research has shown that the fertility recovery of CMS is mostly controlled by a major gene in the nucleus and modified by multiple minor genes [13,14]. The research on Ogura CMS restoration genes is the most in-depth. Delourme and Eber (1992) [15], Delourme (1994) [16], and others selected an isozyme marker, pgi-2 (genetic distance 0.25 cm), closely linked to *Rfo* and obtained several RAPD markers closely linked to *Rfo*, which laid the foundation for later cloning of restoration genes. Brown et al. (2003) [17] and Desloire et al. (2003) [18] used map-based cloning and comparative genomics methods to clone the Ogura CMS fertility restoration gene *Rfo* in radish for the first time. There are currently two main views on the mechanism of action of restoring genes: firstly, the products encoded by restoring genes compensate for the protein products that cannot be expressed normally due to mitochondrial gene recombination, leading to the restoration of plant fertility [19]. The second view is that the restoring gene inhibits the expression of ORFs related to sterile genes, eliminating or weakening the influence of toxic proteins produced by sterile genes. CMS restoration genes can regulate expression at various levels to restore CMS fertility. Bellaoui et al. [20] compared the number of transcripts and the translation efficiency of ORF138 between the Ogura CMS sterile line and the restorer line containing the *Rfo* restoring gene. There was no significant difference between the two, but the ORF138 protein yield in the restorer line was lower than that in the sterile line. Bellaoui et al. [20] believed that *Rfo* was regulated at the post-translational level to restore fertility, and the presence of the restoring gene could reduce the ORF138 protein yield. Except for the aldehyde dehydrogenase encoded by corn Rf2, the reported restoring genes in various crops encode PPR proteins composed of PPR motifs in series. PPR is one of the largest gene families in higher plants. It plays an important role in mediating the expression of plant organelle genes and can regulate the expression of chloroplast and chloroplast genes or RNA splicing, editing, processing, and translation [21].

Chinese cabbage (*B. rapa* L.) originated in China and is a crop of the *Brassica* genus in the Cruciferae family. It is also one of the most important vegetable crops in China. Its annual planting area is about 40 million acres, accounting for about 15% of the total planting area of vegetables in China, making it the largest vegetable crop in terms of cultivation area in China [22]. Cruciferous vegetables are one of the most important vegetables in the world and have important economic value. Therefore, innovation in respect of Chinese cabbage germplasm is of great significance for global food security and social stability. The breeding and utilization of cytoplasmic male sterility in Chinese cabbage began in the late 1970s, but it has not been directly found in Chinese cabbage so far. Therefore, cytoplasmic male sterility in Chinese cabbage mainly comes from alien plants, such as colza, radish, etc. [23]. *Orf138*, *orf222*, and *orf224* are cytoplasmic genes of Ogura-type radish, Nap-type *B. napus*, and Pol-type *B. napus*, respectively, and *orf263* and *orf288* are cytoplasmic genes of mustard cytoplasmic male sterile types [24]. In recent years, four different cytoplasmic male sterile types have been successfully used as male sterile materials in Chinese cabbage [23]. Ogura CMS is a naturally occurring mutation found in radish that was later transferred to *Brassica* crops *(Brassica oleracea*, *B. napus*, *B. rapa*, and *Brassica juncea*). In recent years, Ogura CMS hybrids have become increasingly popular due to their high seed purity. However, these hybrids cannot be reused because Ogura CMS is maternal and all offspring exhibit MS, which hinders self-pollination and the use of these plants in material innovation. Golden-heart Chinese cabbage is named after its golden inner leaf color, and its nutrient content is very rich, especially in terms of carotenoid types and content, which is superior to ordinary Chinese cabbage. Carotenoids are not only essential for plant growth and development but also play important roles in the human body, such as in cardiovascular protection, cancer prevention, and lung system protection. Due to the inability of animals and humans to synthesize carotenoids themselves, they can only be obtained from plants. Therefore, it is particularly important to innovate regarding the germplasm of golden-heart Chinese cabbage.

Golden-heart Chinese cabbage is an Ogura CMS sterile type and cannot be reused because of its cytoplasmic sterility. Establishing a recovery line for Ogura CMS in Chinese cabbage is an effective way to solve this problem. In this study, *GOLDEN* gene linkage markers were used to screen Ogura CMS germplasm with golden hearts and hybridize them with canola containing the *Rfo* gene as the male parent to restore their fertility. Individuals with good fertility and containing the *GOLDEN* gene were selected as male parents based on molecular marker-assisted breeding (MAS), floral organ development, and pollen viability measurements and hybridized with high-quality rapeseed materials. Plants were selected with normal cytoplasm and *GOLDEN* genes in their offspring but without recovery genes. The successful application of canola containing *Rfo* to Ogura CMS golden-heart Chinese cabbage has enabled it to successfully restore its fertility without relying on restoring genes. The results promote the breeding of the *GOLDEN* gene and the reuse of germplasm resources in golden inner leaves in Chinese cabbage.

## 2. Materials and Methods

### 2.1. Plant Materials and Growth Conditions

Two commercial varieties of Chinese cabbage were named F544 and F545, respectively. They are Chinese cabbages with golden-yellow inner leaves. They belong to the Ogura CMS sterile type and served as female parents. Four fertile *B. napus* accessions (M540, M541, M542, and M543) were used as paternal parents. Among them, M540, M542, and M543 contained the fertility restoration gene (*Rfo*), and M541 as a control did not contain *Rfo*. Three high-quality *B. rapa* (cabbage heart material) were also used, named F5577, F3898, and F5270. All the above plants were grown in a glass greenhouse at the Institute of Vegetables and Flowers (IVF) of the Chinese Academy of Agricultural Sciences (CAAS).

### 2.2. Population Construction and Genetic Efficiency

The population construction process is illustrated in Figure 1. The F1 population was obtained by crossing Ogura CMS golden-yellow heart Chinese cabbage as the female parent and canola as the paternal parent. Individual F1 plants that met the target traits were selected as the male parents. Three good portions of Chinese flowering cabbage materials, which contained neither golden yellow inner leaves nor recovery genes and had normal cytoplasm, were used as the female parent. Crosses were performed again to obtain offspring populations. Individual plants that fit the target traits were screened in the offspring population; these plants were the intermediate material for the fertility recovery of golden-heart Chinese cabbage.

All crosses were conducted through artificial pollination during the bud stage, and offspring were screened using *GOLDEN* and *Rfo* gene-specific markers. The transmission efficiencies of the *GOLDEN* gene and *Rfo* gene were calculated as follows:Gene transfer efficiency (%) = number of positive plants/total number of individual plants in the population × 100(1)

### 2.3. Morphological Observation of Intermediate Materials

The morphological characteristics investigated included plant type, leaf shape, leaf color, flower color, etc. The research was carried out according to the standards described in the Specification and Data Standard for Chinese Cabbage Germplasm Resources.

### 2.4. DNA Extraction and PCR Amplification

Genomic DNA was extracted using the TPS [25] method. This method requires minimal and inexpensive reagents for the genomic DNA extraction process, and the entire extraction process is non-toxic and pollution-free. It is a resource-saving and environmentally friendly extraction method suitable for DNA extraction from large batches of samples. The DNA concentration was measured using a NanoDrod 1000 spectrophotometer, and the DNA was diluted to a working concentration of 100–200 ng/μL. Samples were stored at −20 °C.

The polymerase chain reaction (PCR) mixture (10 μL) contained 5 μL of 2× Rapid Taq Master Mix (containing Taq DNA polymerase, dNTP, and Mg^2+^), 0.2 µL of forward primers, 0.2 µL of reverse primers, 1 μL of template DNA (100–200 ng μL^−1^), and 3.6 μL of distilled H_2_O. The PCR conditions were as follows: 95 °C pre-denaturation for 5 min; denatured at 94 °C for 30 s, annealed at 60 °C for 30 s, and extended at 72 °C for 30 s over 35 cycles; extended at 72 °C for 5 min; maintained at 4 °C. After PCR amplification, the markers were separated on a 1.5% agarose gel in 0.5× Tris acetate EDTA (TAE) buffer at 180 V for 15 min and visualized under ultraviolet (UV) light.

### 2.5. Fertility Determination

The fertility-related characteristics of *GOLDEN* positive and *Rfo* positive strains were determined, including flower organ development and pollen vitality. Flower organ development was observed directly, and stamen development was mainly recorded. The determination of pollen vitality was carried out using the acetic acid magenta method. Acetic acid magenta dye has good adsorption and affinity abilities. Pollen vitality was measured during the peak flowering period. Three fresh flowers were selected from each plant, followed by shaking off the pollen onto a glass slide, staining with acetic acid magenta dye for 5 min, and observing under an optical microscope. Energetic pollen has a round shape and smooth, non-wrinkled surrounding cells, showing a purple-red color. Each slide records three repeated fields of view, and each repeated field of view shows more than 100 pollen grains as valid replicates under a microscope.
Pollen vitality (%) = number of active pollen grains/total pollen grains × 100(2)

### 2.6. Specific Markers and Genetic Efficiency

Populations were screened using molecular marker-assisted breeding. The specific primers BnRfo-AS2F/BnRfo-AS2R [26] were used to screen out the single strain containing *Rfo*, and the specific primers Bio17F/Bio17R1/Bio17R2 [27] were used to screen out the single strain containing the *GOLDEN* gene (see Appendix A).

### 2.7. Statistical Analysis

The data were analyzed, and all statistical tests were run using R 4.2.0. One-way ANOVA and LSD were used to analyze all experimental data and the mean SD.

## 3. Results

### 3.1. Genetic Regularity of Rfo in F1

#### 3.1.1. The Performance of Fertility Recovery

Two female and four male materials were interspecifically crossed, resulting in a total of eight different combinations. According to Table 1, only six of the eight hybrids succeeded in producing seeds. Two combinations, F544 × M541 and F545 × M542, were pollinated at the bud stage, but no seeds were obtained. The combination with seeds was sown, and the affinity between parents was judged by the plant survival rate; the lower the plant survival rate, the worse the affinity between parents of the combination. The highest survival rate of M103 was 90%, and the lowest survival rate of M105 was 64.29%. The average survival rate of the six combinations was 78.01%.

In terms of fertility recovery only, the combination with M540 and M542 as the paternal parents was fully restored, but the combination with M541 as the paternal parent was not restored. Interestingly, segregation appeared in the combination fertility recovery with M543 as the paternal parent. After the chi-squared test, the ratio of fertile-sterile segregation conformed to 1:1 at the 0.05 level. In conclusion, the paternal M540, M541, and M542 loci were homozygous, while M543 might be heterozygous. 

#### 3.1.2. Availability of Specific Primers

A marker specific for the *Rfo* gene in *B. napus*, BnRfo-AS2F/R, was used for validation in both parents. The results show that there was no band in the female parent (Chinese cabbage), and there was a band of 40p in the male parent containing the *Rfo* gene. Both M540 and M542 contained *Rfo*, M541 did not, and some of M543 contained *Rfo* and some did not (Figure 2). This is consistent with the fertility recovery in Section 3.1.1, which indicates that material M543 is heterozygous. The progeny population was screened using the Rfo-specific marker BnRfo-AS2F/BnRfo-AS2R, and the effective number of plants was 385, of which 289 were *Rfo*-positive single plants (Table 1). The results were confirmed by field investigation, and the cytoplasm type of all progenies was the Ogura CMS type identified via the orf138-specific marker. The average rate of *Rfo* gene transfer was 67.62% among the six crosses, and the highest rate of *Rfo* gene transfer was 100% in crosses M101 and M104. One cross, M105, had zero *Rfo* transmission, and no *Rfo*-positive strains were screened.

#### 3.1.3. Morphological Differences between Offspring and Parents

The plant type, leaf shape, and flower organs were very similar among the female parents. The same was true for the four paternal materials. Therefore, any male/female material was selected as a schematic for comparison with the offspring. There were six different combinations of F1, but there was little difference in the three aspects of plant type, leaf shape, and flower organs through morphological observation. Either combination was selected for comparison of morphological differences with the parent version (Figure 3). There was little difference in plant type among the six crosses, and the overall morphology belonged to the intermediate type. The plant body was large, and the main stem was strong, forming more branches and exhibiting strong branching, which was more evenly distributed in each stem segment of the main stem. The branches were long in the lower part and became shorter closer to the top (Figure 3). Elongated basal stems and leaves showed obvious linear defects; moss stems and leaves were lanceolate, without linear defects. The moss stems and leaves were semi-enclosed, which is in line with the salient common characteristics of canola (Figure 3). In addition, the leaf size was closer to that of the father. From the perspective of floral organs, the floral organs of F1 belong to the intermediate type, and the distance between flowers is neither shorter nor longer than that of the mother and father; the flower density is between that of the parents (Figure 3). It can be clearly seen from Figure 3 that the flowers of the mother parent are small and the flowers of the father parent are large. The size of the flowers of the offspring and the angle between the petals belong to the intermediate type. The color of the parents is slightly different, and the color of the father is brighter than that of the mother. The colors of F1 appear to be separated; some are between the colors of the parents, some are the same as the parents, and some are the same as the mothers. Due to the difference in light, the difference in color cannot be seen in Figure 3. That is, it is not easy to judge the color in the photo, and the difference in color can only be seen by direct observation. Except for M105, which did not show fertile floral organs, all other crosses showed fertile floral organs.

#### 3.1.4. Comparison of F1 Pollen Vitality

To investigate whether there are significant differences in fertility between different hybrid combinations, further research was conducted on flower organs. According to Figure 4A, the six hybrid combinations have the same flower size and color (due to different lighting conditions in the picture). From the perspective of stamens (Figure 4B), sterile plants have short stamen filaments, long anther cones that are flattened and even degenerated into filaments, and do not have pollen. The development of stigma and stamens in fertile plants was normal, and stamens were dispersed normally. Pollen vitality was measured. M103 had the highest pollen vitality of 68.64%, while M102 had the lowest pollen vitality of 54.22%. M105 had no fertile plants. An analysis of variance was conducted on the pollen viability of both the father and the offspring, and the results show a *p*-value of 2.04 × 10^−15^, less than 0.001, indicating that the pollen vitality varied among 10 different materials and that the difference was statistically significant. From the perspective of effect quantity, the larger the effect quantity, the stronger the relationship between the independent variable and the dependent variable. The proportion of the total error in pollen vitality values explained by different materials was as high as 88.85%. From Figure 4D, we can see that there was no significant difference in pollen vitality among the four male parent materials. There were significant differences in pollen vitality between offspring, except for the control group M105 and the other five groups. M101 and M104 come from the same male parent, which was M540. The male parent of M103 and M106 was the same as that of M543, but there was no significant difference in pollen vitality between M101 and M104 or between M103 and M106, indicating that the female parent does not affect the pollen vitality of the offspring. The female parents of M101, M102, and M103 were the same, but there was a significant difference in pollen vitality between M101, M10, and M102, and the difference between M101 and M103 was not significant. In terms of male parents, M540 and M543 had no significant impact on the pollen vitality of their offspring, and the results were not significant. However, M542 had a greater impact on the pollen vitality of its offspring. The female parent of M104, M105, and M106 was the same as that of F545. There was a significant difference between M104, M106, and M105, and there was no significant difference between M104 and M106. F540 and F543 had no significant effect on the pollen vitality of offspring, and the results were not significant, consistent with the above results. F541 does not contain the restoring gene *Rfo*, and its offspring are not fertile, which has a significant impact on the pollen vitality of the offspring.

### 3.2. Genetic Rule of GOLDEN Gene in F1

#### 3.2.1. Characteristics of the Female Parent

The selected female parent materials in this study were the imported golden-heart Chinese cabbage. Unlike the common Chinese cabbage in the market, which has a white heart in the inner leaves, the color of the inner leaves in the golden-heart Chinese cabbage is yellow (Figure 5A), hence the name. Orf138F/R, orf222F/R, and orf224F/R [28] are specific primers of cytoplasmic male sterility genes that are used to identify the cytoplasmic genes of maternal infertility. As can be seen from Figure 5B, only *orf138* fragment bands were amplified from the two female parent materials. The results show that the two female golden-heart Chinese cabbage samples were indeed of the Ogura CMS sterile type.

#### 3.2.2. Genetic Rule of *GOLDEN* Gene in F1

Zhang et al. [27] designed the co-separation marker Bio17, which can be used as a molecular marker for the identification of golden-heart Chinese cabbage. The specific marker Bio17 for the *GOLDEN* gene in Chinese cabbage was validated in both parents, and the results are shown in Figure 5C. The target bands of 606 bp (Bio17F and Bio17R1) and 268 bp (Bio17F and Bio17R2) were obtained in the female parent, but two bands could not be amplified in the four male parents.

As shown in Table 2, a total of 483 plants from the progeny population were screened using the *GOLDEN*-specific marker Bio17, and 244 *GOLDEN*-positive individuals were obtained. Among the six crosses, the average transfer rate of the *GOLDEN* gene was 51.25%, and the highest transfer rate of M104 was 81.48%. The lowest transmission rate of the hybrid M105 was 33.33%. The *GOLDEN* gene is controlled by a dominant nuclear gene, and its inheritance conforms to Mendel’s law of separation and free combination of independent genes [27]. The desired ratio was 1:1. After the chi-squared test, at the 0.05 level, M101, M102, M103, and M106 were in accordance with the desired proportion. M104 did not meet the chi-square test, but it was much higher than the desired proportion.

### 3.3. Selection of Individual Plants for Traits

Molecular marker-assisted breeding was used to obtain the final target individual for the F1 population. First, 244 *GOLDEN*-gene-positive individuals were selected from 483 plants using the specific primer Bio17. The 244 *GOLDEN*-positive individuals were selected with the specific primer BnRfo-AS2F/R, and 114 individuals with the final target traits (containing both *Rfo* and *GOLDEN*) were finally selected. Combined with phenotypic investigation, the target traits selected were all fertile per plant. The final number of individual plants selected for each combination is shown in Table 2. Among them, M105, the control group, met the expected proportion. M104 was slightly higher than the expected proportion, and the other combinations were lower than the expected proportion. This may be due to the fact that parental influences lead to a decrease in the overall proportion.

### 3.4. Creation of Chinese Cabbage Germplasm with Normal Cytoplasm and a Golden Heart

Among the 114 plants with the final target traits, three individuals with the highest pollen vitality were selected as the male parent, and high-quality *B. rapa* materials were used as the female parent for the second hybridization to obtain the offspring population, creating a golden-heart Chinese cabbage germplasm with normal cytoplasm.

#### 3.4.1. Investigation of Agronomic Traits in the Offspring of Secondary Hybridization

In the early stages, the offspring population was screened using molecular marker-assisted breeding, and a total of 143 plants with the desired traits were screened. Twenty-five robust seedlings free of pests and diseases were selected to colonize the glass greenhouse, and their agronomic traits were investigated at the bolting flowering stage. Through investigation, the offspring groups showed little overall difference in plant type. From the perspective of leaves, the leaf shape of the offspring began to separate, and some of the plants had lanceolar stems and leaves without linear defects. The mossy stem leaves are semi-phimose, which is consistent with the morphological characteristics of the paternal leaves (M6803 in Figure 6B). The leaves of some plants are oval or ovoid, with blunt or slightly concave leaf tips; the leaf base is attached to the stem and the whole margin is intact, which is in line with the morphological characteristics of the leaves of the female parent (M6802 in Figure 6B). Some of the leaves belong to the intermediate type. An example is M6801 in Figure 6B. Alternatively, the offspring groups showed subtle differences in leaf color. Some leaves were darker, and some were lighter. All offspring had normal floral organ development and were able to disperse pollen normally.

#### 3.4.2. Genetic Efficiency of *GOLDEN* and *Rfo* Genes

At the seedling stage, molecular marker-assisted breeding was used to screen a total of 800 offspring populations using *GOLDEN*-specific markers Bio17F/R1/R2 and *Rfo*-specific markers BnRfo-AS2F/BnRfo-AS2R (Figure 6C). As shown in Table 3, a total of 251 *GOLDEN*-positive individuals and 193 *Rfo*-positive individuals were obtained. They accounted for 31.38% and 24.13% of the total, respectively. In terms of the proportion of *GOLDEN*-positive single plants, the proportion of M6803 was the highest, reaching 68.42%. From the single *Rfo*-positive strain, M6802 was the highest; its proportion was 33.14%. In terms of the proportion of the desired target traits (containing *GOLDEN* but not *Rfo*) per plant, M6802 had the highest proportion, followed by M6803 and M6801.

#### 3.4.3. Creation of Normal Cytoplasmic Germplasm

Through molecular marker-assisted breeding, screening of *GOLDEN*-positive plants and *Rfo*-positive plants for restoring genes in offspring populations was undertaken. Based on the investigation of agronomic traits (The overall morphology of the plant is closer to that of *B. rapa*), one combination (M6802) ultimately resulted in a single plant that contains the *GOLDEN* gene but does not contain the *Rfo* gene (the red box marked in Figure 6C is the result of marker screening for this individual plant). The field investigation results show that the plant was fertile and had an overall morphology that deviated from the female parent (Figure 7). Finally, high-quality intermediate material for restoring the fertility of Ogura CMS golden-heart Chinese cabbage germplasm was obtained.

## 4. Discussion

### 4.1. Creating Ogura CMS Recovery Material by Distant Hybridization

The cytoplasmic male sterility of radish belongs to the nucleocytoplasmic interaction type. The classical genetic analysis of traits is based on equal gamete activity. Due to the genotype difference in female and male gamete activity, the accuracy of the genetic analysis of some traits is often affected [29]. The Ogura CMS cytoplasmic male sterile system of radish has stable and complete fertility and is considered to be the most ideal cytoplasmic male sterile type for the heterosis utilization of *Brassica* crops [30]. However, when the restoring gene is transferred from radish, the redundant linked chromosome fragments also infiltrate into the rape genome and have a certain impact on the genetic characteristics of the restoring gene [29]. Previous studies have suggested that the fertility restoration of radish CMS is controlled by at least two independent dominant genes. Further research found that the fertility of Ogura CMS can be restored by the regulation of the *Rfo* gene via the mitochondrial chimeric gene *orf138* and the nuclear fertility-restoring gene *Rfo*. At present, research on the fertility restoration of the Ogura CMS type is sparse in respect of cruciferous crops and more in-depth in cabbage vegetables such as kale [31], cabbage [32,33,34], and broccoli [35], and there is no systematic research on cabbage. In 2022, Zhang et al. [27] found the dominant gene controlling the golden-heart character in Chinese cabbage for the first time. The golden-heart Chinese cabbage is rich in carotenoids, has a prominent color and obvious characteristics, and has high research value. Due to cytoplasmic male sterility, there has been no system for the study of golden-heart Chinese cabbage. If the fertility of Ogura CMS golden-heart Chinese cabbage is restored, this will provide direct materials and a scientific basis for the reuse and germplasm innovation of high-quality Ogura CMS golden-heart Chinese cabbage.

### 4.2. Selection of F1 Material

In order to expedite the breeding process, the method of molecular marker-assisted breeding was used to select positive plants containing the *GOLDEN* gene at the F1 seedling stage and retain the positive plants in order to carry out green-body vernalization, so as to transform the vegetative growth into reproductive growth and enable bolting and flowering as soon as possible. The fertility of the positive plants was observed at the flowering stage, and the agronomic traits were investigated. The results show that the progenies of the combination with M541 as the male parent were completely sterile. With M543 as the male parent, the fertility of the combination was separated, and the separation ratio reached 1:1. The offspring of the combination with M540 as the male parent were all fertile. The above results again verify that the restoring gene *Rfo* can restore the fertility of Ogura CMS. Interestingly, the offspring from the combination with M542 as the male parent should be fertile. However, during the fertility investigation, it was found that the two sterile plants were somewhat different from the ideal test results. It was speculated that they might have been artificially pollinated at the bud stage or that the *Rfo* gene contained in the male parent material had some changes, which could be further studied.

### 4.3. The Significance of the Bridge Material in Chinese Flowering Cabbage

In order to obtain the fertility restoration germplasm of Ogura CMS golden-heart Chinese cabbage, the male parent of canola containing the restoration gene was selected to obtain the offspring. Chinese flowering cabbage prefers a mild climate, has wide adaptability, and has no strict requirements with regard to the photoperiod. It can be cultivated all year. It has the advantages of a short growth cycle, low-temperature requirements for bolting development, normal bolting, and flowering even without low-temperature induction in production, rapid seed-setting, and a large number of seeds. In order to speed up the breeding process, high-quality cabbage material was selected as the male parent for the second crossing, and its offspring can quickly bolt and blossom without vernalization. This is unlike the mechanism of “early bolting” caused by the low temperature of typical cruciferous vegetables such as Chinese cabbage and cabbage, which can greatly shorten the breeding time and provide new ideas for the rapid innovation of its germplasm.

### 4.4. Possibly the Most Suitable Strain as a Restorer in F1

Comparing all the results, since there was no significant difference in morphological observation among the six hybrid combinations, the offspring were selected from four aspects: the affinity between parents, pollen viability, *Rfo* gene transfer rate, and whether the offspring contained both *Rfo* and *GOLDEN* genes. M105, as a blank control group, did not participate in the selection. In terms of the affinity between parents, M103 has the highest ratio of the six hybrid combinations. In terms of pollen viability, M103 had the highest average pollen viability. The highest rate of *Rfo* gene transfer was 100% in crosses M101 and M104. Only M104 achieved the expected ratio in terms of whether it contained both genes. In summary, M103 and M104 are two more promising potential strains for restoring fertility to Ogura CMS *B. rapa* among the six hybrid combinations. However, it should be noted that the single plant we finally screened was from M6802. The male parent of M6802 also happens to be M103. Taken together, we believe that the most appropriate and optimal restorer line may be M103.

## 5. Conclusions

In this study, a distant cross between *B. rapa* with the golden heart trait and *B. napus* containing the *Rfo* gene was performed. Combined with morphological observation, pollen viability measurement, and MAS, 114 individuals with restored fertility and the golden heart trait were screened. Three individuals (containing both *Rfo* and *GOLDEN* genes) with the highest pollen viability were used as the male parents for the second cross. *B. rapa* with normal cytoplasm was used as the female parent. By the same means, one plant with restored fertility but no *Rfo* gene, normal cytoplasm, and the golden heart trait was obtained.

## Figures and Tables

**Figure 1 genes-14-01613-f001:**
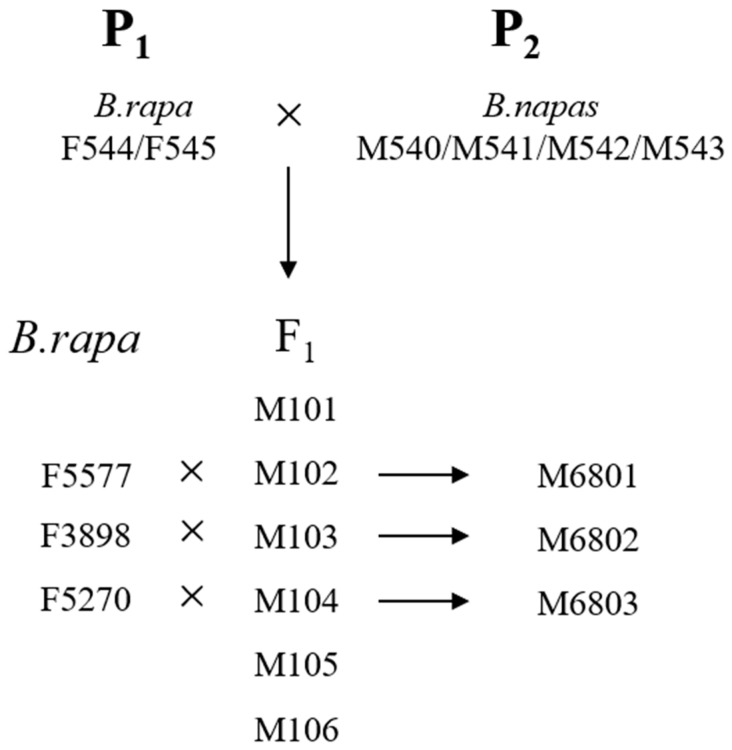
The process of group construction.

**Figure 2 genes-14-01613-f002:**
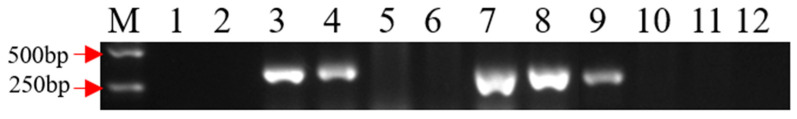
Testing of the availability of the specific marker BnRfo in both parents. M: marker 2000, 1: F544, 2: F545, 3–4: M540, 5–6: M541, 7–8: M542, 9–11: M543, 12: blank control group.

**Figure 3 genes-14-01613-f003:**
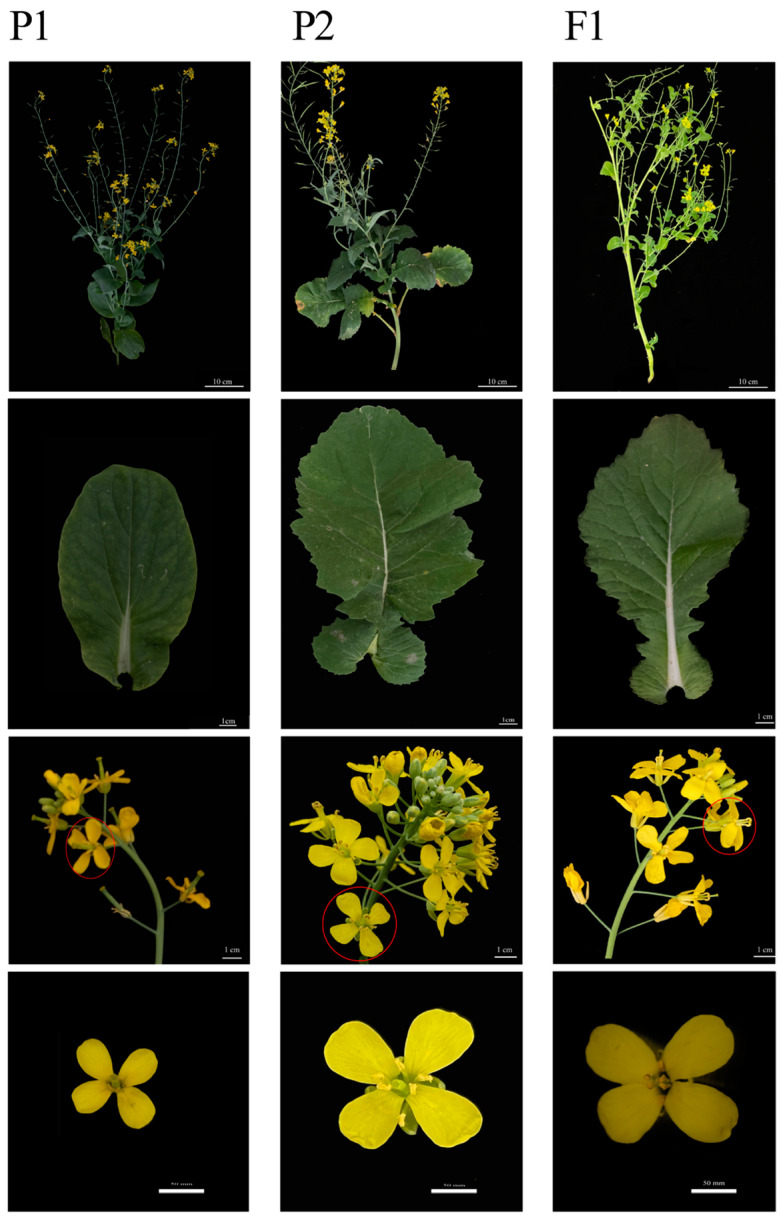
Comparison of agronomic traits between parents and offspring.

**Figure 4 genes-14-01613-f004:**
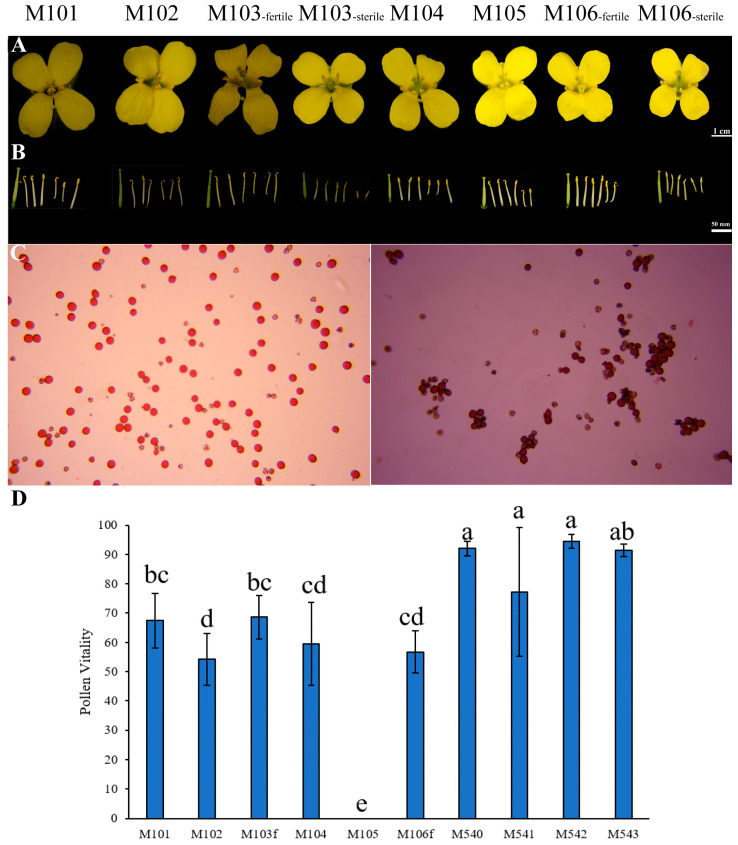
Further exploration of the floral organs of F1. (**A**) Comparison of flower organs in six different combinations of hybrid progeny. (**B**) Dissected flower organs for further observation of stamens and stigma. (**C**) On the left is the offspring M103 with the highest pollen vitality, while on the right is the offspring M102 with the lowest pollen vitality. Pollen vitality measurements were conducted under a 100× optical microscope. (**D**) Analysis of significant differences in pollen vitality between male parents and F1. Mean ± SD.

**Figure 5 genes-14-01613-f005:**
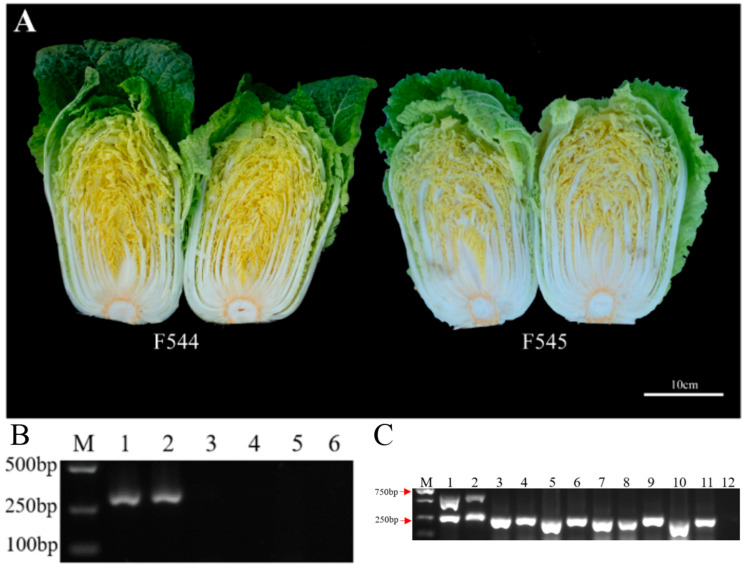
Characteristics of the female parent. (**A**) Longitudinal cut of F544 and F545; the golden yellow inner leaves are visible. (**B**) PCR amplification using the cytoplasmic male sterility gene-specific markers orf138F/orf138R, orf222F/orf222R, and orf224F/orf224R in 2 Chinese cabbage samples. CMS type identifications of F544 and F545. 1–2: primer orf138F/R; 3–4: primer orf222F/R; 5–6: primer orf224F/R. F544 for lanes 1, 3, and 5; F545 for lanes 2, 4, and 6. (**C**) PCR amplification using the *GOLDEN*-specific marker Bio17 in the parents. 1: F544, 2: F545, 3–4: M540, 5–6: M541, 7–8: M542, 9–11: M543, 12: blank control group.

**Figure 6 genes-14-01613-f006:**
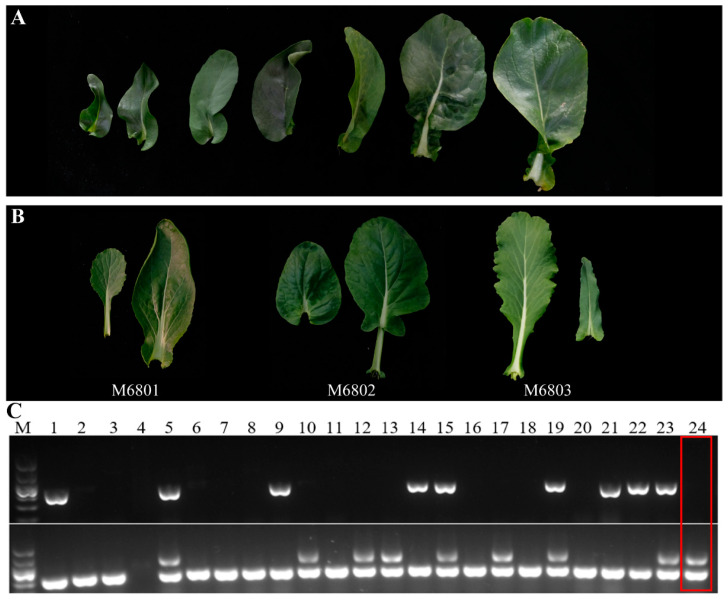
A survey of population traits in the offspring. (**A**) Schematic diagram of the leaf morphology of common Chinese flowering cabbage material. (**B**) Observation of leaf morphology in the offspring. (**C**) The results of the offspring gel map were screened using molecular markers (partial results). The marker BnRfo-AS2F/R was used in the upper gel plot, and the marker Bio17 was used in the lower gel plot. The size of the marker was 200p. Lanes 1–24 are all single plants of the progeny population that were screened.

**Figure 7 genes-14-01613-f007:**
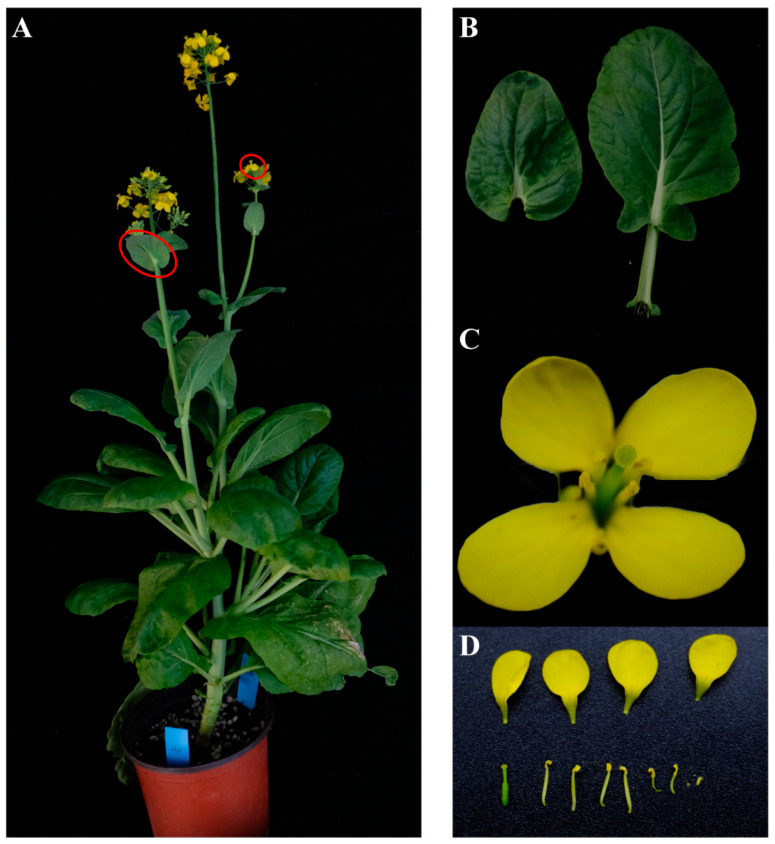
Ogura CMS is a high-quality intermediate material for restoring the fertility of golden-heart Chinese cabbage germplasm. (**A**) Overall morphological characteristics of the intermediate material. (**B**) Intermediate material leaf shape. (**C**) Flowers with intermediate materials. (**D**) Anatomical diagram of the flower of the intermediate material.

**Table 1 genes-14-01613-t001:** Investigation of F1 plant survival rate and fertility.

P_1_	P_2_	F_1_	Number of Seeds	Number of Ungerminated Seeds	Number of Surviving Plants	Survival Rate of F_1_ (%)	Number of Fertile Plants	Number of Sterile Plants	Tested Ratio	χ^2 1^
F544	M540	M101	128	24	104	81.25	90	0	—	—
	M541	—	0	—	—	—	—	—	—	—
	M542	M102	130	26	104	80.00	70	0	—	—
	M543	M103	130	13	117	90.00	56	39	1:1	3.04
F545	M540	M104	80	26	54	67.50	44	0	—	—
	M541	M105	56	20	36	64.29	0	24	—	—
	M542	—	0	—	—	—	—	—	—	—
	M543	M106	80	12	68	85.00	29	33	1:1	1.47

^1^ χ^2^_0.05_ (1) = 3.84.

**Table 2 genes-14-01613-t002:** Number of positive individual plants containing target genes screened for each combination and transmission rate of genes in F1.

F_1_	Combination	Total Number of Investigated Plants	Number of *GOLDEN*-Positive Plants	Transmission Rate of *GOLDEN* in F_1_ (%)	Tested Ratio	χ^2 1^	Number of Positive Plants Containing Both *Rfo* and *GOLDEN*	Proportion of Expectation	Actual Proportion	Difference Value
M101	F544 × M540	104	49	47.12		0.35	34	0.5	0.33	0.17
M102	F544 × M542	104	49	47.12		0.35	25	0.5	0.24	0.26
M103	F544 × M543	117	55	47.01		0.42	15	0.25	0.13	0.12
M104	F545 × M540	54	44	81.48		21.41	29	0.5	0.54	−0.04
M105	F545 × M541	36	12	33.33		4.00	0	0	0	0
M106	F545 × M543	68	35	51.47	1:1	0.06	11	0.25	0.16	0.09

^1^ χ^2^_0.05_ (1) = 3.84.

**Table 3 genes-14-01613-t003:** Number of positive individual plants containing target genes screened for each combination and their proportion of all the plants.

Name	Combination	Total Number of Plants	Number of *GOLDEN*-Positive Plants	Actual Proportion (%)	Number of *Rfo*-Positive Plants	Actual Proportion (%)	Number of Plants Containing *GOLDEN* but Not *Rfo*	Actual Proportion (%)
M6801	F5577 × M102	612	169	27.61	131	21.41	94	15.36
M6802	F3898 × M103	169	69	40.83	56	33.14	45	26.63
M6803	F5270 × M104	19	13	68.42	6	31.58	5	26.32
Total		800	251	31.38	193	24.13	144	0.18

## Data Availability

The data presented in this study are available upon request from the corresponding author.

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
