# Peer review of "Development of Ogura CMS Fertility-Restored Interspecific Hybrids for Use in Cytoplasm Replacement of Golden-Heart Materials in Brassica rapa"

_genes, 2023, doi:10.3390/genes14081613_

Round 1

Reviewer 1 Report

The reviewed paper focuses on diversity of Brassica crops. This work is well-written and properly designed. However, I have two major comments which in my opinion should be taken into account:

(1) Authors did not define the purpose of this work and did not include conclusions

(2) Figure 4D should be verified - vigor means vitality (%)? In M&Ms section there is information about vitality and from Fig 4D vitality ± SD or SE (we do not know) for M541 is over 100% How is it possible? Furthermore, letters for statistics are confusing (should be improve due to the rule that subsequent letters are for decreasing values.

Minor comment:

(1) Table 2 and 3 present show partly the same results, I recommend merging them into 1 table

(2) I found two typos in Abstract: line 15 'Brassica napus containing Rfo gene was used' and line 19 'GOLDEN' not 'GOLDED'. Please verify entire manuscript.

Author Response

Point-by-point response to reviewer 1

Dear reviewer:

Thank you very much for your careful investigation throughout the whole paper. We had revised the manuscript according to your advice. Additionally, we have made further language changes to the paper. Your critique is very important to our paper. Thank you again for your assistance; it is very much appreciated.

Comment 1:  Authors did not define the purpose of this work and did not include conclusions.

Response: CMS was widely used in seeds production in Brasscia rapa crops. But it also restricted germplasm utilization. Rfo gene can restore CMS into fertile, but there was no Rfo gene in Brassica rapa. In this research, our purpose was using Rfo gene in Brassica napus to restore Brassica rapa’s CMS with distant crosses. And we successfully transferred CMS Brassica rapa with GOLDEN gene in to fertile materials. We revised the whole manuscript, and added CONCLUSION in lines 538-546.

Comment 2: Figure 4D should be verified - vigor means vitality (%)? In M&Ms section there is information about vitality and from Fig 4D vitality ± SD or SE (we do not know) for M541 is over 100% How is it possible? Furthermore, letters for statistics are confusing (should be improve due to the rule that subsequent letters are for decreasing values.

Response: We re-examined the data and performed an ANOVA. The letter marking method was also adjusted accordingly. Fig 4D vitality mean ± SD. This was changed in lines 334-340.

Comment 3: Table 2 and 3 present show partly the same results, I recommend merging them into 1 table.

Response: We accept your suggestion to merge Tables 2 and 3 into one table and name it Table 2. This was changed in line 380.

Comment 4: I found two typos in Abstract: line 15 'Brassica napus containing Rfo gene was used' and line 19 'GOLDEN' not 'GOLDED'. Please verify entire manuscript.

Response: The correct way to write it is GOLDEN. We corrected this error and double-checked the entire manuscript. And ABSTRACT was revised in lines 13-29.

Reviewer 2 Report

Manuscript ID genes-2519275

Li et al., submitted the manuscript entitled “Development of Ogura CMS fertility-restored interspecific hybrids for use in cytoplasm replacement of golden-heart materials in Brassica rapa “for publication consideration in Genes.

This study provided a study of Brassica napus that contained Rfo gene which was used to transfer the golden heart trait from Ogura CMS to fertile materials in Brassica rapa by distant crossing. Morphological and molecular marker assisted selection methods were used to screen the individuals in each generation. A total of 483 individuals (F1) were obtained from the interspecific cross between Ogura CMS golden-heart Brassica rapa and Brassica napus with Rfo gene. The manuscript provided valuable information for increased the diversity of Brassica rapa germplasm and provided a new method for the utilization of CMS germplasm in Brassica cropsThe research is well planned and carried out. 

The scientific soundness of this manuscript is acceptable and I think it meets the aims and scope of Genes journal. 

Authors are encouraged to consider following points:

1.     Please carefully check through the writing, some language issues occur. 

For example, affiliations labeling is not corresponding. 

ZL1, Guoliang Li, Fei Li, Shifan Zhang, Xiaowu Wang, Jian Wu, Rifei Sun, Shujiang Zhang * and Hui Zhang * 

Institute of Vegetables and Flowers, Chinese Academy of Agricultural Sciences, Beijing, China, Sciences, Beijing, China,  State Key Laboratory of Vegetable Biobreeding, Institute of Vegetables and Flowers, Chinese Academy of  Agricultural Sciences, Beijing, China 8 13163135502@163.com (Z.L.); liguoliang@caas.cn (G.L.); lifei@caas.cn (F.L.); zhangshifan@caas.cn (S.Z.); wangxiaowu@caas.cn(X.W.); wujian@caas.cn(J.W.); sunrifei@caas.cn (R.S.)  Correspondence: zhangshujiang@caas.cn (S.Z.); zhanghui05@caas.cn (H.Z.)  

Line 102, 103,  “Vegetables are the crops with the widest cultivation area in China and the most important economic status besides other food crops.” . This sentence is not well-written.

2.     Abstract section could be organized in a way to easy understandable, current writing is hard to follow. 

 Overall, authors presented an importance and sound report. 

Round 2

Reviewer 2 Report

Authors have addressed issues.